# Tuning of Magnetic Hyperthermia Response in the Systems Containing Magnetosomes

**DOI:** 10.3390/molecules27175605

**Published:** 2022-08-31

**Authors:** Matus Molcan, Andrzej Skumiel, Milan Timko, Ivo Safarik, Kristina Zolochevska, Peter Kopcansky

**Affiliations:** 1Institute of Experimental Physics, Slovak Academy of Sciences, Watsonova 47, 04001 Kosice, Slovakia; 2Faculty of Physics, Adam Mickiewicz University, Uniwersytetu Poznańskiego 2, 61-614 Poznań, Poland; 3Department of Nanobiotechnology, Biology Centre, ISBB, CAS, Na Sadkach 7, 370 05 Ceske Budejovice, Czech Republic; 4Regional Centre of Advanced Technologies and Materials, Czech Advanced Technology and Research Institute, Palacky University, Slechtitelu 27, 783 71 Olomouc, Czech Republic

**Keywords:** alternating magnetic field, rotating magnetic field, magnetic nanoparticles, magnetic hyperthermia, heat evolution

## Abstract

A number of materials are studied in the field of magnetic hyperthermia. In general, the most promising ones appear to be iron oxide particle nanosystems. This is also indicated in some clinical trial studies where iron-based oxides were used. On the other hand, the type of material itself provides a number of variations on how to tune hyperthermia indicators. In this paper, magnetite nanoparticles in various forms were analyzed. The nanoparticles differed in the core size as well as in the form of their arrangement. The arrangement was determined by the nature of the surfactant. The individual particles were covered chemically by dextran; in the case of chain-like particles, they were encapsulated naturally in a lipid bilayer. It was shown that in the case of chain-like nanoparticles, except for relaxation, a contribution from magnetic hysteresis to the heating process also appears. The influence of the chosen methodology of magnetic field generation was also analyzed. In addition, the influence of the chosen methodology of magnetic field generation was analyzed. The application of a rotating magnetic field was shown to be more efficient in generating heat than the application of an alternating magnetic field. However, the degree of efficiency depended on the arrangement of the magnetite nanoparticles. The difference in the efficiency of the rotating magnetic field versus the alternating magnetic field was much more pronounced for individual nanoparticles (in the form of a magnetic fluid) than for systems containing chain nanoparticles (magnetosomes and a mix of magnetic fluid with magnetosomes in a ratio 1:1).

## 1. Introduction

Iron-based magnetic nanoparticles are often investigated especially in the field of bioresearch. In particular, increasing attention is being paid to research on magnetic hyperthermia. The principle follows from the magnetic nanoparticles’ ability to induce local heating under the influence of various types of applied alternating magnetic fields (AMF) [1]. It is extremely difficult to determine the ideal procedures for the preparation of a sample for hyperthermia as well as the method of its characterization. It is necessary to take into account a number of parameters, such as material, equipment, and the selection of a suitable measuring or application methodology. Hyperthermic indicators such as the heating rate or maximum power output are affected by particle concentration, particle size and shape, type of surfactant, viscosity of the surrounding medium, etc. [2]. The concentration of magnetic nanoparticles (NPs) has a great effect on hyperthermia properties, as shown by Fe_3_O_4_ NPs at various concentrations when treated in an alternative magnetic field; it was observed that the Δ*T* sharply increases with increasing the NPs concentration while the specific absorption rate (SAR) remains almost constant [3,4]. Moreover, the presence of interparticle interactions can influence the hyperthermia properties of magnetic NPs in an AC magnetic field [5]. Surface coating of magnetic nanoparticles can have a substantial effect on the magnetothermal properties, as shown in the case of CoFe_2_O_4_ magnetic nanoparticles modified by triethylene glycol (TEG) coating, which caused increased NPs saturation magnetization as measured by SQUID. It was shown that TEG coating increases the heating efficiency of the CoFe_2_O_4_ magnetic NPs due to an increase in saturation magnetization and a decrease in the strength of the magnetic interactions between the coated nanoparticles [6]. In addition, the bimagnetic NPs can exhibit more interesting properties than single-phase NPs, as shown in the case of cube-like bimagnetic hard/soft (CoFe_2_O_4_/Fe_3_O_4_) and soft/hard (Fe_3_O_4_/CoFe_2_O_4_) nanocomposites (core/coating) with an average dimension of 20 nm. The CoFe_2_O_4_/Fe_3_O_4_ nanocomposites presented a larger saturation magnetization than the CoFe_2_O_4_ NPs, which is effective for their potential use in magnetic hyperthermia [7]. In general, the basic requirement is to achieve maximum heating of a unit amount of magnetic substance per unit time under the conditions of the applied magnetic field at a given frequency. This efficiency is known as the Specific Absorption Rate (SAR) [8]. Of course, input parameters such as frequency and intensity of the applied magnetic field have a significant effect on the heating effect [9]. On the other hand, the limitations/limits of magnetic hyperthermia must not be forgotten in order to avoid unwanted and dangerous patient discomfort [10]. The most well-known criterion is the so-called Brezovich’s criterion [11], stating that the AMF is harmless to the human body if its amplitude *H*_0_ and frequency *f* satisfy the condition *f**·H*_0_ < 4.85 × 10^8^ A∙m^−1^∙s^−1^). Apart from Brezovich’s criterion, others can also be found in the literature. Hergt et al. [12] suggested a less rigid criterion where *f*·*H* = 5 × 10^9^ A·m^−^^1^·s^−^^1^. When examining the heating effect, we must also take into account the method of generating the magnetic field, which can significantly affect the efficiency [13]. The most commonly used method in the field of magnetic hyperthermia is to generate an AMF. The mechanisms underlying the production of heat for nanoparticles include the Néel mode (rotation of the magnetic moment of the magnetic nanoparticles) and rotational friction provided by the viscous drag of the suspending medium as the nanoparticles align with the magnetic field [14]. However, studies appear where the possibility of increasing efficiency by means of a so-called rotating magnetic field (RMF) exists [15,16,17,18,19]. The RMF is AMF, which creates a resultant field during the superposition of two or more AMFs of identical frequency but spatially displaced in phase with respect to each other [19]. The motivation of this work was to analyze the heating effect of magnetite nanoparticles of different configurations by two approaches. Both AMF and RMP were used to heat them. The response of particles in the form of stable colloid of individual particles (magnetic fluid), in the form of chains (magnetosomes), as well as a mix of colloidal systems of magnetic fluid and magnetosomes, was monitored.

## 2. Materials and Methods

The magnetic hyperthermia effect was studied at three types of magnetite colloids of different characters. The first sample represented the individual magnetite nanoparticles in the form of dextran stabilized magnetic fluid. The second sample was magnetite magnetosome nanoparticles in the chain-like structure form, and the last sample represented the mix of colloidal systems of magnetic fluid and magnetosome chains. For better illustration, magnetosomes are chain-like structures of a single domain, single crystal magnetic nanoparticles of magnetite, Fe_3_O_4_ or greigite, Fe_3_S_4_ (depending on the species of bacteria), enclosed and connected by a lipid bilayer membrane [20]. The magnetosome chains are formed by the biomineralization process [21] in magnetotactic bacteria.

In order to determine the hydrodynamic size of the prepared samples, the dynamic light scattering measurements (DLS) were carried out on a Malvern Zetasizer Nano ZS (Malvern, UK) as well as magnetization measurements (VSM magnetometer—Cryogenic Ltd., London, UK) to determine core size distribution function.

### 2.1. Samples Description

#### 2.1.1. Individual NPs (Magnetic Fluid)

Dextran stabilized Fe_3_O_4_ magnetic nanoparticles were synthetized using the modified Molday procedure [22]. The dextran (70 kDa, Sigma Aldrich; 19 g) was dissolved in water (75 mL), while FeCl_3_·6H_2_O (5 g) and FeCl_2_·4H_2_O (2.1 g) were dissolved in 2 M HCl (13 mL). The solutions were mixed and placed in a water bath (60 °C). Subsequently, under mixing, 7.5% ammonium hydroxide (75 mL) was added dropwise, and the mixing continued for another 15 min at 60 °C. The next day the magnetic fluid was centrifuged (5000 rpm, 45 min; Universal 320, Hettich Zentrifugen, Germany) [23].

#### 2.1.2. Chain-Like NPs (Magnetosomes)

Magnetosomes as a product of *AMB-1* magnetotactic bacteria were prepared under laboratory conditions. The culture conditions and magnetosome extraction recipe were described in detail recently [24]. Extracted magnetosome nanoparticles were dispersed in the HEPES buffer.

#### 2.1.3. Colloidal Systems of Magnetic Fluid and Magnetosomes

The sample was prepared by mixing the magnetic fluid and the magnetosome suspension in a ratio of 1:1, (1 mL of magnetosomes suspension, *φ*_v_ = 0.019% + 1 mL of Dextran FF, *φ*_v_ = 0.29%).

### 2.2. Apparatus for Generating a Magnetic Field

#### 2.2.1. Rotating Magnetic Field

The test samples were exposed to both rotating and oscillating high-frequency magnetic fields. Due to the low concentration of nanoparticles in the samples (from *φ*_v_ = 0.019% for magnetosomes to 0.29% for dextran magnetic fluid) and thus due to the expected low efficiency of the calorimetric effect, systems were developed to record and measure temperature changes during operation magnetic field (Figure 1). In the case of RMF, a modernized version of the system described in more detail in our previous articles was used [17,18]. The main change to the circuit was the use of three transformers with ferrite cores to raise the voltage supplying parallel resonant circuits (*L*_P_ coils and *C*_P_ capacitors). The tested sample was located in the central part of the magnetic circuit, where three magnetic fluxes shifted in phase and space by 120 angular degrees and were flowing successively. Due to the superposition, a rotating magnetic field was created. In the case of AMF, the magnetic field was linearly polarized.

#### 2.2.2. Alternating Magnetic Field

For AMF generation, a double-layer coil (Figure 2) with self-inductance L = 29.6 μH was used, which was connected in parallel with a polypropylene capacitor with an electric capacity of C = 64 or 54 nF to obtain the frequency *f* = 115 or 126 kHz. The first layer of the coil was a thin-walled copper tube, and the second layer was a copper wire consisting of many wires with a total area of S_Cu_ = 2 mm^2^. The electric voltage across the resistor R= 0.1 Ω was used to measure the current flowing through the coil. During the calibration of the system (before the actual measurement of the calorimetric effect), a measuring coil (voltage probe) was inserted into the center of the coil. The measurement of the amplitude value of the magnetic field intensity *H* was based on the registration of the voltage induced in the probe’s coil with the surface SC at the frequency *f*, which allowed one to calculate the value H using the basic laws of electromagnetic induction (i.e., Faraday’s law). The parallel LC circuit was powered by the 300 W Power Amplifiers AL-300-HF. Its maximum values of current and voltage amplitudes are I = 15.1 [A_p_] and U = 39.6 [V_p_]. Thanks to the use of a ferrite transformer on the secondary winding, the voltage amplitude U = 252 [V_p_] and the corresponding amplitude of the current consumed by the parallel circuit I = 1.68 [A_p_] were obtained. According to additional research carried out by the authors, in a parallel LC circuit, during resonance, the current of the coil and the capacitor is several dozen times greater than the current supplying this circuit. Both these currents are 180 degrees shifted, and this is the most important advantage of a parallel LC system compared to a series one. In both RMF and AMF systems, the sample temperature was measured with an optical fiber sensor. In experiments, the optical fiber temperature sensor [13] by FISO Technology Inc., model FOT-L-SD, was used with a temperature range of −40 °C to 300 °C, with a response time better than 1.5 s, accuracy of 0.1 °C, and resolution of 0.01 °C.

## 3. Experimental Part

### 3.1. Nanoparticle Systems Characterization

The hydrodynamic size distribution performed by DLS is shown in Figure 3. In the case of the magnetic fluid sample (black line), we can see a uniform peak. Relatively width peak is a result of dextran coating. The magnetosome sample (green line) exhibits bimodal size distribution (observed before [25]) because of the polydispersity of magnetosome chains. The polydispersity is caused by various numbers of particles per chain [26] and various shapes of isolated chains as a result of their natural ability loops creation after isolation [27]. The mixture of magnetic fluid and magnetosomes (red line) represents a dominant peak (170 nm), which is attributed to signal from the dextran magnetic fluid particles, and a peak with a lower intensity (970 nm) attributed to magnetosomes. These size results point to different sample characteristics that may lead to different responses to the applied magnetic fields.

Magnetization curves under static DC field conditions (presented in Figure 4a) show superparamagnetic behavior. The differences in magnetization saturation are caused by different concentrations of the magnetic component (magnetite) in the analyzed sample. When the scale is changed (Figure 4b), a small coercivity (and hysteresis) in the case of a magnetosome sample can be visible. In order to prove it more clearly, it should be measured with a more sensitive device (for example, SQUID) and with a higher density of measured points. Moreover, such a kind of magnetization is a result of the high shape anisotropy of magnetosomes, and it was experimentally observed also in other works [28,29]. The dynamic of magnetization increase (magnetic susceptibility slope), which leads to fast saturation, depends on the magnetite concentration but also can be attributed to size distribution. In the case of smaller and more uniform particles (magnetic fluid sample), we can see a faster response on the applied field compared with the mixed sample and magnetosome sample (Figure 4b).

Additionally, the magnetic core diameter of tested samples was calculated by fitting the initial magnetization curve to the model of magnetization in polydisperse magnetic fluid. The particle size distribution, in this case, is usually described by the log-normal distribution function [30]:(1)$$f\left(d\right)=\frac{1}{\sqrt{2\pi d\beta }}exp\left[\frac{-{\left(ln\frac{d}{d_{0}}\right)}^{2}}{2\beta^{2}}\right]$$
where *d* is the diameter, *d*_0_ and *β* are the parameters obtained by fitting this function. The magnetite nanoparticle distribution functions are shown in Figure 5. In order to estimate the mean diameter ⟨d⟩ and mean standard deviation ⟨σ⟩, the following formulas were used [31]: $d=d_{0}exp\left(\frac{\beta^{2}}{2}\right)$ and $\sigma =d\sqrt{exp\left(\beta^{2}\right)-1}$.

In the case of Dextran FF magnetite nanoparticles, the mean size was estimated to be 16.8 nm. The magnetosomes magnetite cores were estimated to be 35.6 nm. The magnetosomes core size is comparable to the study of Gojzewski et al. [32]. The obtained mean size and its standard deviation were 43 ± 12 nm. However, it should be noted that, in this case, it is not individual nanoparticles but particles arranged in chain-like structures. Electron microscopy images of MTS are not presented now, but they can be seen, for example, in the following papers [24,25,32,33].

### 3.2. Heating Effect Results in RMF

The results of the rotating magnetic field’s influence on the tested samples are shown in Figure 6, Figure 7 and Figure 8. Temperature changes before and after switching on the RMF for subsequent samples are shown in Figure 6a, Figure 7a and Figure 8a. The obtained values of temperature changes proved to be small due to the low concentration of magnetic material in the samples (from 0.019% in the sample of MTS to 0.29% in the sample Dextran FF) and did not exceed 100 mK in the experiment. In order to determine the temperature slope in time (d*T*/d*t*), the linear function was fitted to the recorded temperature measurement points in time after switching on the RMF.

In turn, Figure 6b, Figure 7b and Figure 8b show the dependence of (d*T*/d*t*) on the value of the RMF intensity amplitude *H*. In the case of the Dextran FF sample, the measurement points (d*T*/d*t*)–*H* are arranged along the power function (d*T*/d*t*) = (*H*/2068)^1.97^. For the remaining tested samples, the measuring points are arranged in a manner similar to the linear relationship.

As shown in Figure 6b, in the magnetic fluid sample of dextran-coated nanoparticles, there is mainly a relaxation mechanism of heat energy release according to the Néel mechanism. This fact is proved by the value of the exponent *n* = 1.97, close to 2, which is characteristic of superparamagnetic nanoparticles lacking magnetic hysteresis [30].

### 3.3. Heating Effect Results in AMF

The AFM calorimetric effect results at similar frequencies are shown in Figure 9, Figure 10 and Figure 11. Figure 9a, Figure 10a and Figure 11a show the temperature changes of the samples before and after switching on the magnetic field. Due to the achieved higher AMF intensity values (over 10 kA/m), higher temperature increases were obtained. In this case, the thermal fluctuations did not matter much.

The dependence of the measurement points of the *dT/dt* parameter on the AMF intensity *H*, shown in the Figure 9b, Figure 10b and Figure 11b, was supplemented with a power function of the d*T*/d*t* = (*H/a*)*^n^* type, where *a* and *n* are numerical parameters, obtained from the fitting procedure.

The dependence of the power function d*T*/d*t* on the AMF intensity shown in Figure 9b reveals a comparable value of the exponent of this function (*n* ≃ 2) that was obtained in RMF, which proves that the mechanisms of heat released are similar in RMF and AMF.

In the case of the sample with magnetosomes, where the magnetic cores are much larger than that of Dextran FF, it can be seen from Figure 10b that the exponent *n* = 2.37. This means that apart from the magnetic relaxation mechanism, there is also an additional source of heat release, which is magnetic hysteresis. Remembering that in the case of losses due to magnetic hysteresis, the thermal power *P* ∝ (d*T*/d*t*) separated according to this mechanism is proportional to the cube of the AMF amplitude, *P*_his_ ∝ *H*^3^, we can write the resultant power function (*H*/*a*)*^n^* obtained from the matching as:(2)$$\frac{dT}{dt}={\left(\frac{H}{4897}\right)}^{2.37}={\left(\frac{H}{5348}\right)}^{2}+{\left(\frac{H}{8072}\right)}^{3}$$

Formula (2) thus includes two heat sources in a sample of magnetosomes exposed to AMF. At the same time, it can be seen from Figure 10b that the dominant mechanism of the heat released, however, is magnetic relaxation. Then we can formally present the parameter (d*T*/d*t*) as consisting of two effects from relaxation (*H/r*)^2^ and from hysteresis (*H/h*)^3^:(3)$$\frac{dT}{dt}={\left(\frac{H}{a}\right)}^{n}={\left(\frac{H}{r}\right)}^{2}+{\left(\frac{H}{h}\right)}^{3}={\left(\frac{H}{5348}\right)}^{2}+{\left(\frac{H}{8072}\right)}^{3}\;$$

### 3.4. Comparison of the Heating Effects of Samples for Both Types of Magnetic Fields

From the comparative dependences (Figure 12, Figure 13 and Figure 14) of the heating rates (d*T*/d*t*) in RMF and AMF, we can observe much higher heating dynamics (i.e., efficiency) in RMF for all three types of particles (individual, chain-like, and mixed). The most significant difference in the efficacy of RMF to AMF, at comparable *f* values, can be observed in the case of individual nanoparticles of the Dextran FF sample. In the case of systems containing chain-like nanoparticles (MTS and Dextran FF + MTS), the d*T*/d*t* values are always higher when the RMF is applied, but the differences are not as significant as in the case of individual nanoparticles of the Dextran FF sample. These facts correspond to the findings of Bekovic’s works [15,16], where he presented higher heat losses in RMF compared to AMF. Experimentally, this finding is also presented in [17], where, in oil-based magnetic fluid, RMF caused a heating effect of more than twice as large as AMF.

Table 1 provides numerical values of *a*, *n*, *r*, and h parameters obtained by fitting experimental data of magnetite nanoparticle systems’ heat response during the application of RMF and AMF. Having determined *a, n, r,* and *h* coefficients from Equation (3), it is possible to calculate the share of thermal energy from the magnetic hysteresis in relation to the total losses in heating from the Equation (3) [34]:(4)$$\frac{P_{hys}}{P_{total}}=\frac{H\cdot r^{2}}{h^{3}+H\cdot r^{2}}$$

Figure 15 shows this proportion for comparison for the samples with magnetosomes and for the mixed two samples (magnetosomes and Dextran FF). As can be seen in both cases, the source of the loss in magnetic hysteresis is the sample with magnetosomes, but the sample “mix” gives a smaller share of the loss on hysteresis than the sample with the magnetosomes alone, which is obvious. The Dextran FF sample itself showed no hysteresis loss due to its superparamagnetic properties.

## 4. Specific Absorption Rate (SAR) Analysis

The primary goal of the work was not to maximize hyperthermic parameters (d*T*/d*t* and SAR) but to point out the influence of the method of generating the magnetic field (AMF and RMF). On the other hand, the ability of magnetic particle systems to dissipate the energy of an external alternating magnetic field is given by the SAR (W∙kg^−1^; the ratio of the absorbed power to the mass of magnetic particles in the sample) normalized to the unit mass of magnetic material [35]. Moreover, Table 2, Table 3 and Table 4 below summarize the calculated SAR values at the given fields and frequencies, which were used in the measurement of the heating dependencies. In order to calculate SAR, the following formulas and parameters were used:(5)$$\mathrm{SAR}={\;C}_{S}\frac{m_{S}}{m_{\mathrm{NP}}}\cdot \left(\frac{dT}{dt}\right),\;\frac{m_{S}}{m_{\mathrm{NP}}}=\frac{1}{\phi_{V}}\cdot \frac{\rho_{S}}{\rho_{\mathrm{NP}}}$$

C_S_ ~ 4184 [J/kg·K], ρ_NP_ = 5180 [kg·m^−3^], ρ_S_ = 1000 [kg·m^−3^], m_NP_ = Φ_V_·ρ_NP._

**Table 2 molecules-27-05605-t002:** AMF parameters (*H* and *f*) and heating rate (d*T*/d*t*) of tested samples.

Samples (RMF)	*H* [A·m^−1^]	*f* [kHz]	d*T*/d*t* mK·s^−1^	Φ_V_ [%]	Φ_V_ [-]	m_NP_ [kg]	m_S_/m_NP_	SAR [W·kg^−1^]
Dextran FF	1047	122	0.29	0.29	0.0029	15.0	66.7	80.9
	1653	122	0.59					165
	1879	122	0.86					240
MTS	1396	125	0.19	0.019	0.00019	0.984	1016	808
	2569	125	0.36					1530
	3439	125	0.51					2168
Dextran FF + MTS	1438	125	0.23	0.154	0.00154	7.98	125	120.3
	2462	125	0.57					298.1

**Table 3 molecules-27-05605-t003:** RMF parameters (*H* and *f)* and heating rate (d*T*/d*t*) of tested magnetic samples.

Samples (AMF)	*H* [A·m^−1^]	*f* [kHz]	d*T*/d*t* [mK·s^−1^]	Φ_V_ [%]	Φ_V_ [-]	SAR [W·kg^−1^]	ILP = SAR/*f*·*H*^2^ nH·m^2^/kg
Dextran FF	3348	115	0.23	0.29	0.0029	66	0.05
	4756	115	0.595			166	0.06
	6420	115	2.0			558	0.12
	9645	115	3.17			884	0.08
	11,456	115	4.7			1311	0.09
MTS	6732	126	1.67	0.019	0.00019	7103	1.24
	9484	126	5.17			21,988	1.94
	14,092	126	12.17			51,759	2.07
Dextran FF + MTS	3814	126	0.71	0.154	0.00154	373	0.20
	7800	126	3.55			1864	0.24
	11,305	126	5.33			2798	0.17

**Table 4 molecules-27-05605-t004:** Comparison of the heating effect in the tested samples of different physical parameters in an RMF.

Physical Parameter	Symbol	Unit	Dextran FF	MTS	Dextran FF + MTS
Frequency	*f*	Hz	1.22 × 10^5^	1.25 × 10^5^	1.25 × 10^5^
Amplitude	*H*	A·m^−1^	1879	3439	2462
Volume concentration	Φ_V_	-	0.0029	0.00019	(0.00019 + 0.0029)/2 = 0.00154
Heating rate	d*T*/d*t*	K·s^−1^	0.00086	0.00051	0.00057
Specific absorption rate	SAR	W∙kg^−1^	240	2168	299
(d*T*/d*t*)/(Φ_V_·*f·H^2^*)		K·m^2^·A^−2^	6.88 × 10^−13^	18.2 × 10^−13^	4.88 × 10^−13^
ILP = SAR/(*f·H^2^*)		nH·m^2^/kg	0.557	1.47	0.69

Table 4 compares the normalized parameter (d*T*/d*t*)/(Φ_V_·*f*·*H*^2^) [30,36] for the three samples tested in RMF. It was established that the highest value (18.2 × 10^−13^) is achieved by this parameter for the sample from magnetosomes, in which, due to the large size of nanoparticles, there are probably additional power losses caused by magnetic hysteresis. Furthermore, the sample from magnetosomes has the lowest volume concentration value of 0.019%. Another formula in the literature, ILP = SAR/(*f∙H*^2^) [3,4,5,6,7], indicates that our samples give a comparable value of this parameter to the other literature data. However, the magnetosome material provides the greatest thermal effect of any of our samples. It appears that both parameters can be successfully used when comparing different calorimetric effects as long as d*T*/d*t* is proportional to *H*^2^. This is the case when the nanoparticles show a calorimetric effect from magnetic relaxation and, in addition, when the magnetic field is not too high. At higher amplitudes of the magnetic field intensity (when ξ > 1), the nonlinearity effect occurs, and then the power function index *n* < 2.

## 5. Conclusions

The paper shows the influence of the methodology of the application of alternating magnetic fields on the heating of magnetic colloidal systems of various characters. The assumption that a higher heat generation efficiency is achieved in the case of RMF application than in AMF conditions was confirmed. The particle size, as well as the factor of their arrangement (in the form of individual synthetically stabilized nanoparticles, chain-like particles with a natural envelope, or their mix in a ratio of 1:1), also played an important role in the heating mechanism. The heat energy released principle of individual nanoparticles (Dextran FF) originated in the relaxation mechanism according to Néel and Brown relaxation in both RMF and AMF. This is indicated by the value of the exponent *n* exponent close to 2, which is characteristic of superparamagnetic nanoparticles. When the systems contained the magnetosomes (MTS and Dextran FF + MTS 1:1 sample), the power law exponent *n* > 2; in that case, except for magnetic relaxation, a magnetic hysteresis contribution to heating is present. For the Dextran FF sample, the d*T*/d*t* values were significantly different when compared to RMF and AMF. This difference was not so significant for samples containing magnetosomes, although d*T*/d*t* was still higher in the RMF. This suggests that individual nanoparticles have a more pronounced response to RMF than those in chain-like structures. This is probably related to the more dominant contribution originating from the relaxation processes in the hyperthermic experiment. In any case, as already indicated, in magnetic hyperthermia, it is necessary to take into account many factors that affect heat generation. RMF increases the effectiveness of hyperthermia over the often studied AMF. However, higher efficiency is markedly visible in the individual nanoparticle systems of magnetic fluid.

## Figures and Tables

**Figure 1 molecules-27-05605-f001:**
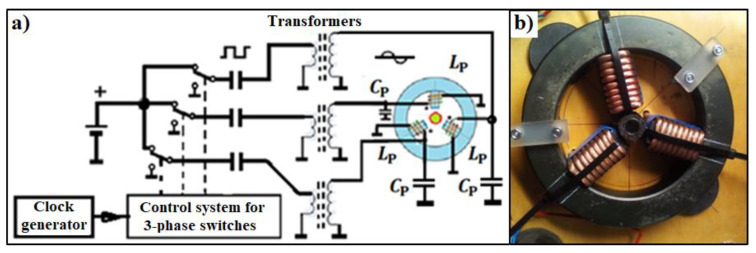
Diagram of a system for generating of rotating magnetic field of high frequency (**a**) and elements of the magnetic circuit: ferrite torus, 3 coils on ferrite cores (**b**).

**Figure 2 molecules-27-05605-f002:**
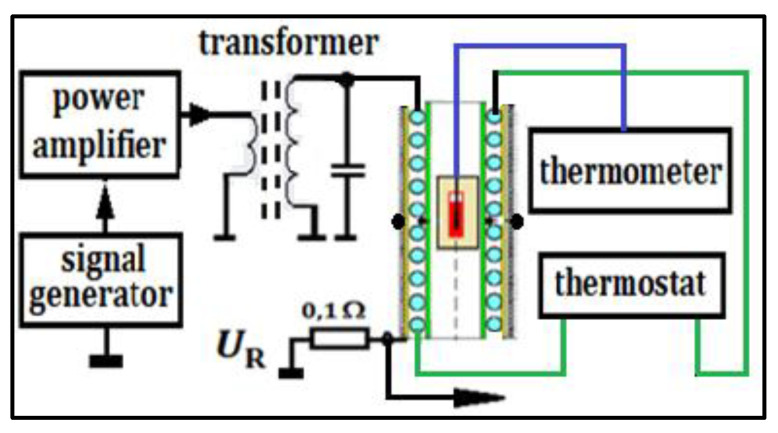
System for generating the alternating magnetic field.

**Figure 3 molecules-27-05605-f003:**
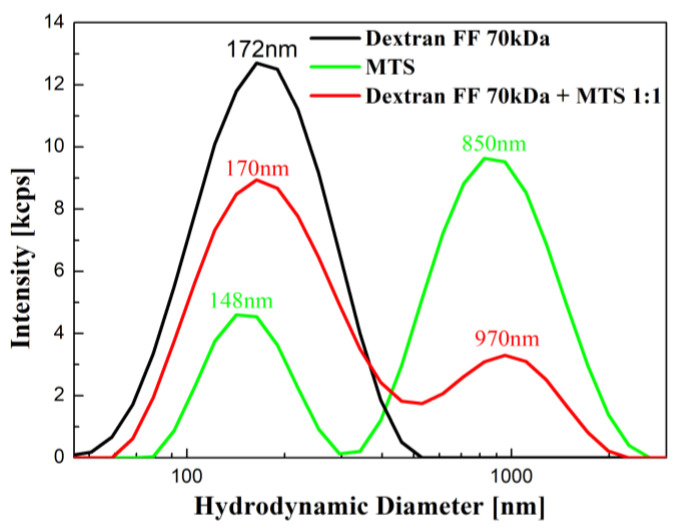
Hydrodynamic diameter distribution of magnetic fluid (dextran-coated Fe_3_O_4_ nanoparticles), magnetosome chains, and the mixture of magnetic fluid and magnetosomes (ratio 1:1).

**Figure 4 molecules-27-05605-f004:**
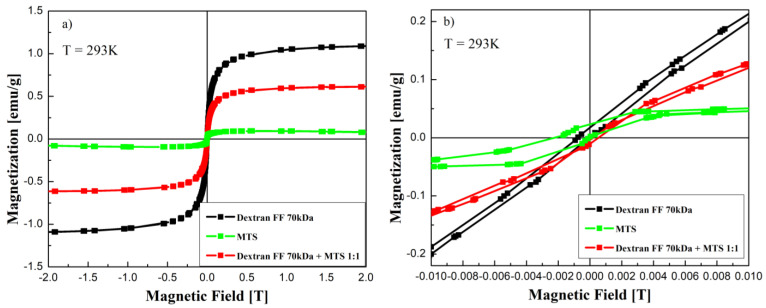
Magnetization versus magnetic field for magnetic fluid, magnetosome chains, and the mixture of magnetic fluid and magnetosomes (ratio 1:1), (**a**) and a zoom resolution where an indication of non-zero coercivity can be observed in the case of magnetosomes (**b**).

**Figure 5 molecules-27-05605-f005:**
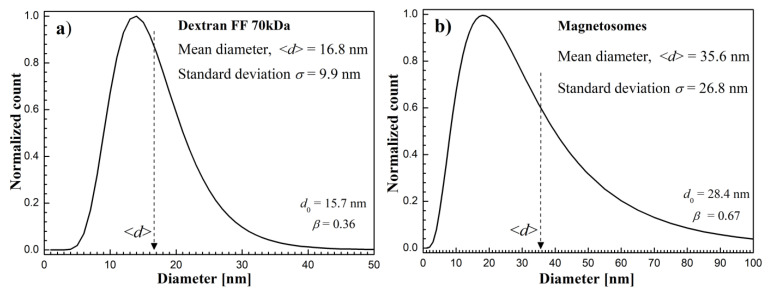
Log-normal particle core size distribution function obtained from VSM data for Dextran FF (**a**) and magnetosomes sample (**b**).

**Figure 6 molecules-27-05605-f006:**
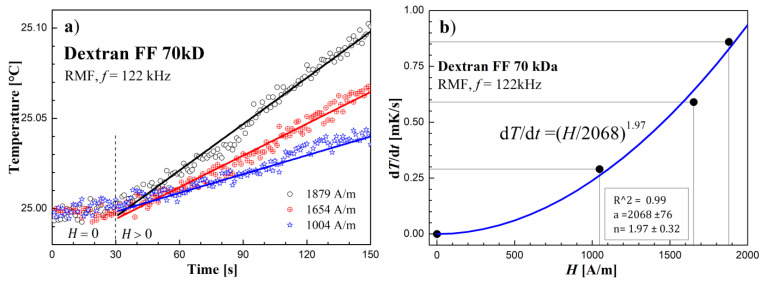
(**a**) Time courses of temperature changes in the Dextran FF sample. (**b**) d*T*/d*t* values depending on the amplitude of the magnetic field strength *H* in the RMF at the frequency *f* = 122 kHz.

**Figure 7 molecules-27-05605-f007:**
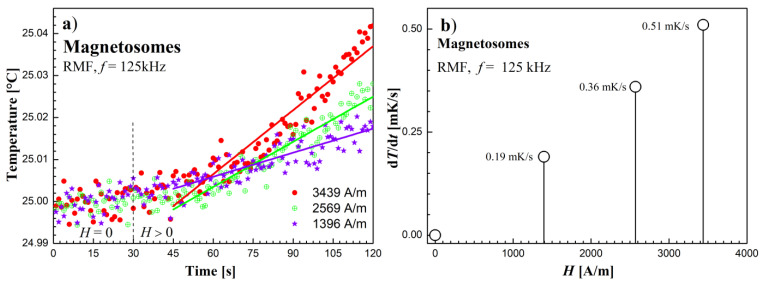
(**a**) Time courses of temperature changes in the magnetosomes sample. (**b**) d*T*/d*t* values depending on the amplitude of the magnetic field strength *H* in the RMF at the frequency *f* = 125 kHz.

**Figure 8 molecules-27-05605-f008:**
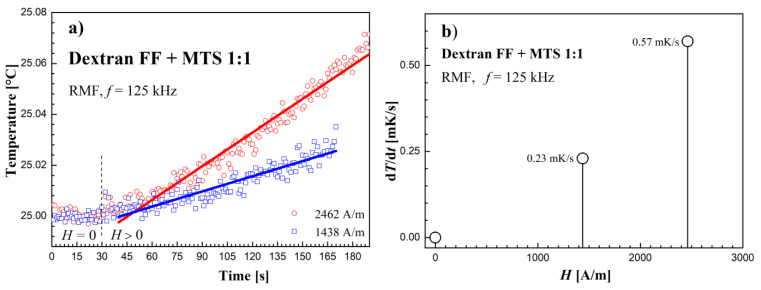
(**a**) Time courses of temperature changes in the dextran with magnetosomes sample. (**b**) d*T*/d*t* values depending on the amplitude of the magnetic field strength *H* in the RMF at the frequency *f* = 125 kHz.

**Figure 9 molecules-27-05605-f009:**
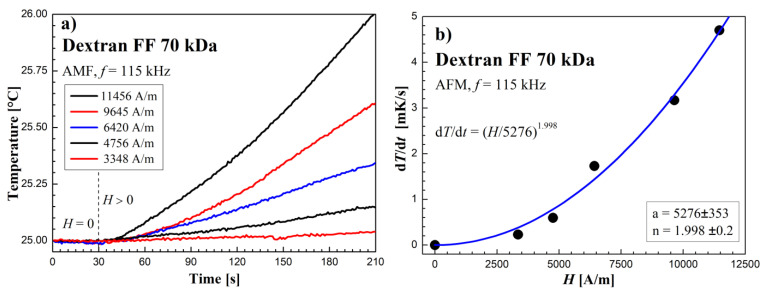
Time courses of temperature changes in the Dextran FF sample (**a**,**b**) d*T*/d*t* values depending on the amplitude of the magnetic field strength *H* in the AMF at the frequency *f* = 115 kHz.

**Figure 10 molecules-27-05605-f010:**
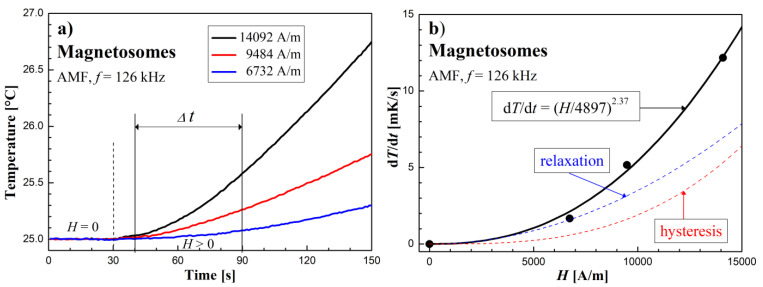
Time courses of temperature changes in the magnetosomes sample (**a**,**b**) d*T*/d*t* values depending on the amplitude of the magnetic field strength *H* in the AMF at the frequency *f* = 126 kHz.

**Figure 11 molecules-27-05605-f011:**
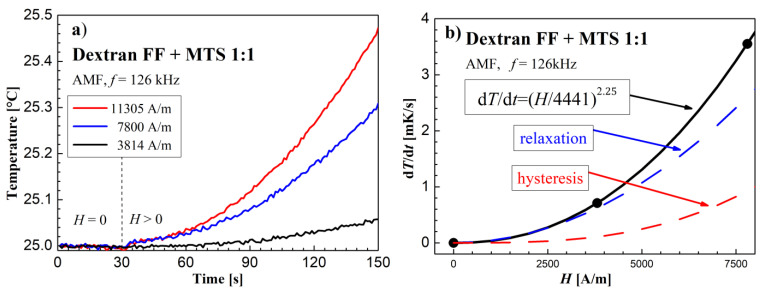
Time courses of temperature changes in the dextran with magnetosomes sample (**a**,**b**) d*T*/d*t* values depending on the amplitude of the magnetic field strength *H* in the AMF at the frequency *f* = 126 kHz.

**Figure 12 molecules-27-05605-f012:**
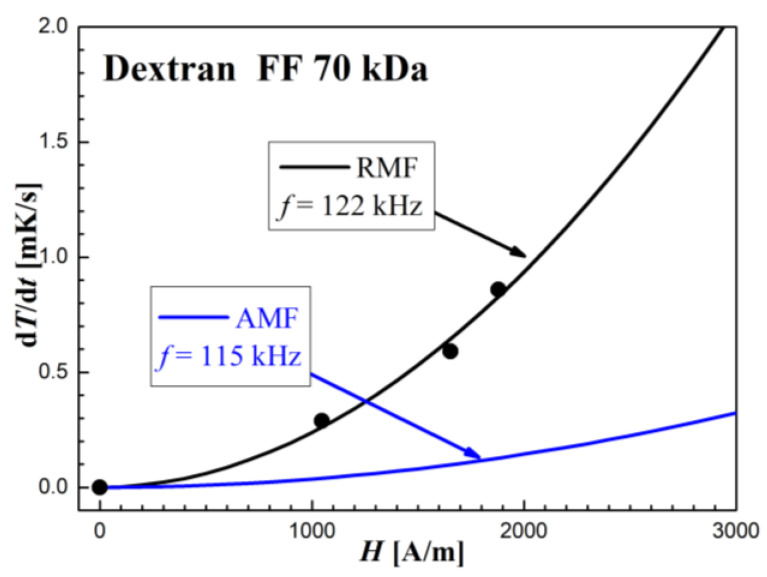
Comparison of the d*T*/d*t* dependence on the amplitude of the intensity *H* measured in rotating and oscillating magnetic fields in the Dextran FF sample.

**Figure 13 molecules-27-05605-f013:**
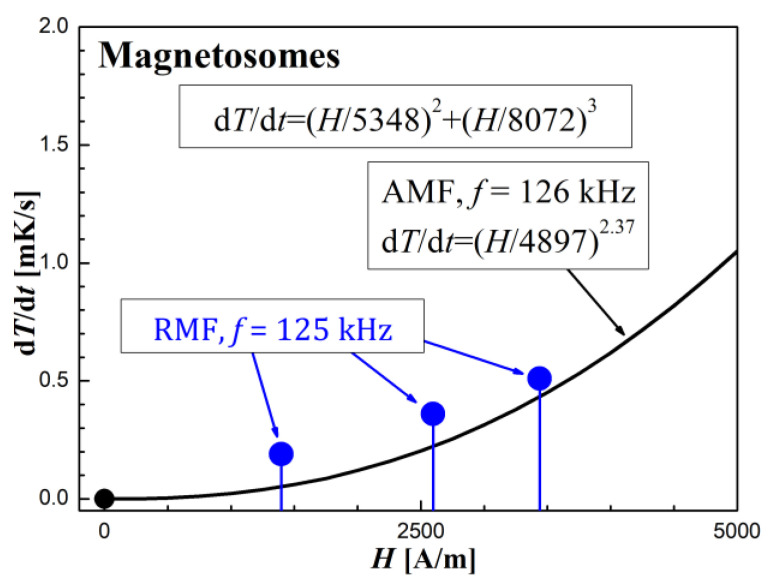
Comparison of the d*T*/d*t* dependence on the amplitude of the intensity *H* measured in rotating and oscillating magnetic field in the Magnetosomes sample.

**Figure 14 molecules-27-05605-f014:**
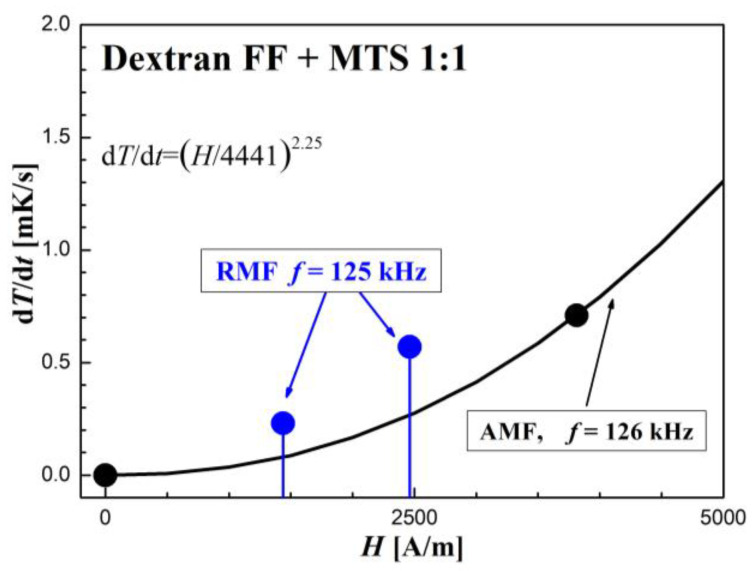
Comparison of the d*T*/d*t* dependence on the amplitude of the intensity *H* measured in rotating and oscillating magnetic field in the sample of Dextran FF + MTS (in a ratio 1:1).

**Figure 15 molecules-27-05605-f015:**
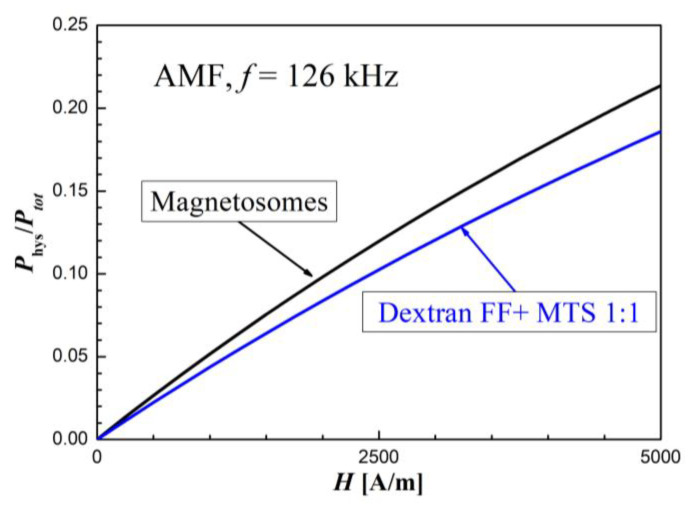
The share of heat energy losses in the magnetosomes sample and in the Dextran FF + MTS sample resulting from magnetic hysteresis in relation to the total loss in a heating effect.

**Table 1 molecules-27-05605-t001:** Numerical values of parameters *a* and *n*, a power function of the d*T*/d*t* = (*H/a*)*^n^* type for samples determined in RMF and in AMF.

Samples	RMF		AMF			
	*a*	*n*	*a*	*n*	*r*	*h*
Dextran FF	2068	1.97	5276	1.998	≅2068	∞
MTS			4897	2.37	5348	8072
Dextran FF + MTS			4441	2.25	4832	7996

## Data Availability

Not applicable.

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
