# Peer review of "Tuning of Magnetic Hyperthermia Response in the Systems Containing Magnetosomes"

_molecules, 2022, doi:10.3390/molecules27175605_

Round 1
Reviewer 1 Report
Dear Editor,
Thank you for considering me as reviewer of the manuscript titled " Tuning of magnetic hyperthermia response in the systems containing magnetosomes".
In this study, the effect alternation magnetic field and rotating magnetic field on hyperthermia output of a system of iron oxide magnetic nanoparticles and chains were reported. There are some unclear point that should be corrected:
· In introduction, please discuss the other effect such as concentration, coating of polymers, aggregation and interparticle interaction on hyperthermia efficiency of ferrite nanoparticles. So please see and use following papers or any other related works:
The effect of interparticle interactions on spin glass and hyperthermia properties of Fe3O4 nanoparticles, Materials Research Express, 4 (2017) 075051.
Superparamagnetic magnetite nanoparticles for cancer cells treatment via magnetic hyperthermia: effect of natural capping agent, particle size and concentration, J. Mater. Sci.: Mater. Electron., 32 (2021) 24026-24040.
Bimagnetic hard/soft and soft/hard ferrite nanocomposites: Structural, magnetic and hyperthermia properties, Ceram. Int., 48 (2022) 4886-4896.
Magnetic hyperthermia properties of iron oxide nanoparticles: The effect of concentration, Physica C, 549 (2018) 119-121.
Magnetic hyperthermia properties of CoFe2O4 nanoparticles: Effect of polymer coating and interparticle interactions, Ceramics Int. (2022).
· If there are other works in which rotating magnetic field is used for hyperthermia please add and compare your results which those.
· In experimental, please add information about the AMF and RMF such as amplitude and frequency and RPM etc.
· The temperature rise after applying the magnetic field is very small even after 120 s. what is the possible reason for such small increase.
· Please calculate the SAR of the samples under different fields and collect the data in a table. Then compare the results with each other and also with the literature. You can use the above mentioned papers in this regard.
With Best Regards.
Reviewer 2 Report
In currently used methods of hyperthermia (general hyperthermia, local hyperthermia using radiofrequency and microwave radiation, ultrasound) the heating of the tumor also causes a significant increase in the temperature of the adjacent healthy tissues. To control the thermal regime, temperature is measured using sensors inserted into the treatment area. Thus, the main disadvantages of the currently used techniques of hyperthermia are low selectivity of exposure, as well as invasive method of temperature control. It is relevant to study magnetic materials in the form of micro- and nanoparticles with unique magnetothermal properties, which can find application in the method of magnetofluid hyperthermia (MFH).
From this point of view the conducted research is relevant.
At the same time, the work raises a lot of questions:
1. What is done new, except, perhaps, the original experimental setup? Coating magnetite nanoparticles with dextrans and studying their various combinations is such an old topic that it does not even make sense to talk about novelty. The authors, by the way, do not provide references to classical works on hyperthermia, but the list cites mostly the authors' own works. For example, most of the parameters affecting the SAR parameter are already discussed here https://aip.scitation.org/doi/abs/10.1063/5.0032843?cookieSet=1
2. The parameters of the experimental installations created are not specified (especially, in the part of the installation for the creation of an alternating magnetic field): field frequency? amplitude? The authors should indicate the range of available values for both experimental setups. Is the Brezovich criterion satisfied?
3. Dependences of temperature on time are not clear at all? Is the heating only up to +25 degrees Celsius? Then what hyperthermia can we talk about? or is it a change in temperature? Then it is obvious overheating?
4. How to interpret Figures 3 and 5? What is the real size of the nanoparticles in the experiment? If it is as large as in figure 3, then we can only speak of a hysteresis heating mechanism at best. If the particles are quite small, is their superparamagnetism considered, as for example here https://www.mdpi.com/2079-4991/11/7/1826. Are the mechanisms described here and the analysis of dT/dt (H) dependences applicable in this case? This requires a separate discussion
The work, as a whole, requires a great deal of revision, with very little physics and modern literature so far
Round 2
Reviewer 1 Report
All the comments were well addressed and the paper is appropriate for publication.
Reviewer 2 Report
All my comments have been addressed